# Safety of Sofosbuvir-Based Direct-Acting Antivirals for Hepatitis C Virus Infection and Direct Oral Anticoagulant Co-Administration

**DOI:** 10.3390/jcm13195807

**Published:** 2024-09-28

**Authors:** Valerio Rosato, Riccardo Nevola, Marcello Dallio, Pierpaolo Di Micco, Angiola Spinetti, Laert Zeneli, Alessia Ciancio, Michele Milella, Piero Colombatto, Giuseppe D’Adamo, Elena Rosselli Del Turco, Paolo Gallo, Andrea Falcomatà, Stella De Nicola, Nicola Pugliese, Roberta D’Ambrosio, Alessandro Soria, Elisa Colella, Alessandro Federico, Maurizia Brunetto, Umberto Vespasiani-Gentilucci, Alessio Aghemo, Pietro Lampertico, Antonio Izzi, Davide Mastrocinque, Ernesto Claar

**Affiliations:** 1Liver Unit, Ospedale Evangelico Betania, 80147 Napoli, Italy; riccardo.nevola@unicampania.it (R.N.); davidemastrocinque4@gmail.com (D.M.); ernestoclaar@gmail.com (E.C.); 2Hepatogastroenterology Division, Department of Precision Medicine, University of Campania “Luigi Vanvitelli”, 80131 Naples, Italy; marcello.dallio@gmail.com (M.D.); alessandro.federico@unicampania.it (A.F.); 3AFO Medica, P.O. Santa Maria delle Grazie, ASL Napoli 2 Nord, 80076 Pozzuoli, Italy; pierpaolo.dimicco@aslnapoli2nord.it; 4Division of Infectious and Tropical Diseases, Department of Clinical and Experimental Sciences, University of Brescia and ASST Spedali Civili, 25123 Brescia, Italy; angiola.spinetti@hotmail.it (A.S.); laert.zeneli@gmail.com (L.Z.); 5Gastro-Hepatoloy Unit, Department of Medical Sciences, University of Turin, 10126 Torino, Italy; alessia.ciancio@unito.it; 6Clinic of Infectious Diseases, University of Bari, University Hospital Policlinico, 70121 Bari, Italy; michele.milella@tin.it; 7Hepatology Unit and Laboratory of Molecular Genetics and Pathology of Hepatitis Viruses, Reference Center of the Tuscany Region for Chronic Liver Disease and Cancer, University Hospital of Pisa, 56124 Pisa, Italy; p.colombatto@ao-pisa.toscana.it (P.C.); maurizia.brunetto@unipi.it (M.B.); 8Umberto I Hospital, UOC General Medicine, 20122 Nocera, Italy; peppedadamo@gmail.com; 9Infectious Disease Unit, Department for Integrated Infectious Risk Management, IRCCS Azienda Ospedaliero, Universitaria di Bologna, 40138 Bologna, Italy; elena.rossellidelturco@aosp.bo.it; 10Clinical Medicine and Hepatology Unit, Fondazione Policlinico Universitario Campus Bio-Medico, 00128 Roma, Italy; paolo.gallo@policlinicocampus.it (P.G.); a.falcomata@unicampus.it (A.F.); u.vespasiani@policlinicocampus.it (U.V.-G.); 11Division of Internal Medicine and Hepatology, Department of Gastroenterology, IRCCS Humanitas Research Hospital, 20089 Rozzano, Italy; stella.denicola@humanitas.it (S.D.N.); nicola.pugliese@humanitas.it (N.P.); alessio.aghemo@gmail.com (A.A.); 12Division of Gastroenterology and Hepatology, Foundation IRCCS Ca’ Granda Ospedale Maggiore Policlinico, 20122 Milan, Italy; roberta.dambrosio@policlinico.mi.it (R.D.); pietro.lampertico@unimi.it (P.L.); 13Clinic of Infectious Diseases, Fondazione IRCCS San Gerardo dei Tintori, 20900 Monza, Italy; alessandroguido.soria@irccs-sangerardo.it (A.S.); elisa.colella@irccs-sangerardo.it (E.C.); 14Research Unit of Clinical Medicine and Hepatology, Department of Medicine and Surgery, Università Campus Bio-Medico di Roma, 00128 Roma, Italy; 15Department of Biomedical Sciences, Humanitas University, 20072 Pieve Emanuele, Italy; 16CRC “A. M. and A. Migliavacca” Center for Liver Disease, Department of Pathophysiology and Transplantation, University of Milan, 20126 Milan, Italy; 17Infectious Diseases, Azienda Ospedaliera D. Cotugno, 80131 Naples, Italy; izziantonio@yahoo.it

**Keywords:** direct oral anticoagulants, direct antiviral agents, drug–drug interactions, hepatitis C, liver cirrhosis

## Abstract

**Background:** Direct oral anticoagulants (DOACs) are recommended for the management of thrombosis prophylaxis, especially in patients with atrial fibrillation. As substrates of cytochrome P450 (CYP) 3A4 and/or P-glycoprotein, they are implicated in potential drug–drug interactions. NS5A/NS5B inhibitors are direct-acting agents (DAAs) against the Hepatitis C Virus (HCV) infection that exert a mild inhibition of P-glycoprotein without effects on CYP3A4. A DOAC and NS5A/NS5B inhibitor co-administration may lead to an increased risk of bleeding. Real-world data on the concomitant use of DOACs and DAAs are scarce. On this purpose, we perform a retrospective analysis on the risk of vascular adverse events (bleeding and thrombosis) among HCV patients under DOAC/DAA therapy, even in advanced liver disease. **Methods:** Between May 2015 and April 2023, patients treated with sofosbuvir-based DAA regimens and DOACs were consecutively included in this study from 12 Italian medical centers. Baseline characteristics, especially concerning bleeding risk and liver function, were collected. The occurrence of bleeding events, classified as major and minor, was the primary endpoint. Secondary endpoints were the rate of any thrombotic events and/or the need for discontinuation of one or both treatments. Moreover, a cohort of patients, matched by demographic characteristics (age and sex), that switched to vitamin K antagonists (VKAs) during the antiviral treatment was compared with the DOAC/DAA group. **Results**: A total of 104 patients were included. Thirty-eight of them (36.5%) were cirrhotic. Atrial fibrillation was an indication for anticoagulation in almost all cases (76%). Rivaroxaban (35.6%) was the most used DOAC, followed by apixaban (26.9%), dabigatran (19.2%) and edoxaban (18.3%). Sofosbuvir/velpatasvir (78.8%) was the most prescribed DAA, and all patients were already on anticoagulant therapy before the start of DAAs. During concomitant DOAC/DAA treatment, no major bleeding events were recorded, while four minor bleeding events occurred, but none resulted in DAA or DOAC discontinuation. At univariate analysis, the only additional risk factor statistically related to bleeding events was the anticoagulant therapy (hazard ratio [HR]: 13.2, 95% confidence interval 1,6-109). Performing an evaluation by a LOGIT binomial analysis with demographic characteristics, the antiplatelet therapy remained statistically associated to bleeding events. No significant differences were found in the rate of clinically relevant bleeding when the main population was compared with the VKA-switched cohort. A single major bleeding event leading to anticoagulation and DAA discontinuation was reported in VKA-switched matched cohort. **Conclusions**: In our study, the concomitant use of NS5A/NS5B inhibitors with DOAC showed good safety, and the only risk factor associated with bleeding events was the concomitant antiplatelet therapy. These findings support the use of DOACs during sofosbuvir-based HCV treatment, even in advanced liver disease. Replacing DOACs with VKAs does not appear to be of clinical benefit.

## 1. Introduction

Although effective and safe therapeutic schemes are now available, chronic HCV infection remains a relevant public health problem, accounting for 56.8 million active infection cases worldwide and causing over 250,000 deaths/year [1]. In the HCV care cascade, the most important step for achieving HCV elimination goals suggested by the World Health Organization (WHO) is represented by the implementation of screening programs [2]. However, there are also still critical issues in the management of treatments with direct-acting antiviral agents (DAAs). These drugs, which are extremely effective and safe, are burdened by some significant limitations resulting from the risk of drug interactions. Most of these drug–drug interactions (DDIs) are linked to metabolism (e.g., cytochrome P450-3 A4, CYP3A4) or hepatic and/or intestinal transporters (e.g., P-glycoprotein, P-gP and breast cancer resistance protein, BCRP) [3]. In particular, potential DDIs have been hypothesized to be among DAAs and direct oral anticoagulants (DOACs).

Due to their ease of handling, DOACs (apixaban, dabigatran, edoxaban, rivaroxaban) have almost completely replaced vitamin K antagonists (VKAs) in the role of first choice drugs for the management of the embolic risk of patients with non-valvular atrial fibrillation or deep vein thrombosis (DVT). As substrates of CYP3A4 and/or P-gP, the enzymatic induction or inhibition by other drugs could be responsible for variations (increasing or decreasing) in the blood concentrations of these anticoagulants, with variable clinical impact [4]. Different from VKAs, there is currently no evidence-based recommendation available for drug concentration measurements, coagulation tests, or target therapeutic ranges that allow for evaluating the intensity of anticoagulation obtained with DOACs [5].

Due to their ability to inhibit P-gP, BCRP or CYP3A4, DAAs may increase DOAC exposure, potentially resulting in increased bleeding risk [6]. For example, preclinical data suggest that the interaction between dabigatran and DAAs results in a significant increase in dabigatran blood concentrations. However, despite the reported increase in DOAC concentrations, there is a paucity of data addressing clinically relevant outcomes such as bleeding. Furthermore, variations in serum concentrations and any clinical effects of interactions between DOACs and DAAs could be affected by the different pharmacokinetic profiles of the drugs used. Apixaban and rivaroxaban are characterized by a predominantly hepatic metabolism (65–75%), mediated at least partially by CYP3A4/5 and CYP2J2, whereas the role of the liver is less significant for edoxaban (50%) and especially for dabigatran (20%) [7]. At the same time, the various antivirals currently used for the treatment of HCV infection also show different pharmacodynamic profiles and, consequently, a different risk of interaction with DOACs. Compared to the combinations of DAAs containing NS5A/NS5B inhibitors (e.g., sofosbuvir/velpatasvir) able to induce only a mild inhibition of P-gP and BCRP, the combinations containing NS3/4A protease inhibitors (e.g., glecaprevir) can induce a stronger inhibition of P-gP and, in addition, a weak inhibition of CYP3A4 [3,6]. This pharmacodynamic profile therefore exposes protease inhibitors to a greater risk of DDIs (Figure 1).

To date, very little post-marketing data are available on the safety of a DAA and DOAC combination. Due to the lack of knowledge of the clinical effects, the concomitant use of these two classes of drugs is currently not recommended or should be used with extreme caution in selected cases [8]. For these reasons, real-world studies appear necessary to verify the clinical effect on the bleeding risk of combinations between DAAs and DOACs.

The aim of this study was to evaluate the incidence of clinically relevant (major and non-major) bleeding during concomitant therapy with DAAs and DOACs (primary end-points) as well as the rate of any thrombotic events and/or the need for discontinuation of one or both treatments (secondary endpoints) and the clinical benefit of switching from DOACs to VKAs during antiviral treatment.

## 2. Material and Methods

### 2.1. Study Design

This was a multicenter, observational, retrospective study that included patients from twelve hepatological centers in Italy. All adult patients with chronic HCV infection who were treated with NS5A/NS5B inhibitors (sofosbuvir/velpatasvir or sofosbuvir/ledipasvir) and DOACs (dabigatran, rivaroxaban, apixaban, edoxaban) for at least 30 days between May 2015 to March 2023 were included. Patients receiving anticoagulant treatment for indications other than atrial fibrillation were also included in this study. All of them had to already be on DOAC treatment before starting antiviral therapy and had to receive at least one clinical follow-up visit 12 weeks after the end or interruption of DAA treatment. We excluded patients with chronic hepatitis B virus and/or human immunodeficiency virus (HIV); active hepatocellular carcinoma; and those on the waiting list for liver transplantation.

At baseline, demographics data, clinical characteristics and comorbidities, DAA and DOAC regimens and indications for anticoagulation were recorded. All characteristics potentially correlated with vascular events (hemorrhagic or thrombotic), including chronic kidney disease, hypertension, stroke or cancer history, alcohol use, concomitant antiplatelet therapy or non-steroidal anti-inflammatory (NSAID) drug use were collected in order to evaluate the HAS-BLED score [9]. The diagnosis of cirrhosis was made according to liver stiffness or histological features, or, if not available, clinical examination, biochemical tests and ultrasonography. Patients with cirrhosis were divided according to the Child–Pugh–Turcotte (CPT) score stage.

Therapeutic regimen and duration of antiviral treatments were defined according to indication of the Italian Drug Agency (AIFA) and international guidelines, including updates that were recommended during this study [8]. The choice of antiviral therapeutic regimen was made by each center according to therapeutic efficacy, patient characteristics and comorbidities and other drug-drug interactions. The achievement of sustained virological response (SVR12) was defined as undetectable HCV-RNA 12 weeks after the end of antiviral therapy.

As a control group, patients that switched to VKAs before the start of DAA treatment were included. The VKA/DAA control group was selected by matching demographic characteristics (age and sex) in a 1:1 ratio with the case group (DOACs/DAAs), in order to obtain comparable populations. The same inclusion and exclusion criteria and the same follow-up methods as the DOAC/DAA cohort were also applied to the control group.

### 2.2. Endpoints

The primary endpoint was the occurrence of bleeding events, classified as major and minor, and of thromboembolic events, during DOAC and DAA co-administration. Major bleeding events, in according to the International Society on Thrombosis and Haemostasis (ISTH) classification, included fatal bleeding, critical site bleeding (intracranial, intraspinal, intraocular, retroperitoneal, intra-articular, pericardial or intra-muscular with compartmental syndrome), gastrointestinal bleeding (i.e., esophageal varices, peptic ulcer) or overt blood loss with a hemoglobin reduction of greater than 2 g/dL from baseline or requiring blood transfusion [10]. Minor bleedings were defined as overt bleeding not meeting criteria for major bleeding but leading to medical intervention, hospitalization or increased level of care prompting face-to-face evaluation.

Secondary endpoints were the rate of DOAC discontinuation due to bleeding or thromboembolic events during or within three months after the completion of DAA treatment.

Moreover, a further objective of this study was the evaluation of the clinical benefit (rate of bleeding or thromboembolism) of switching from DOACs to VKAs during antiviral treatment.

Finally, the rate of SVR 12 was analyzed between the study cohorts.

### 2.3. Statistical Analysis

Continuous variables are expressed as mean values and standard deviations, and categorical data are represented as frequencies and percentages. The Student t-test and the Mann–Whitney U test were performed to compare differences in the values of continuous variables. Fisher’s exact test and the chi-square test with the Yates correction were used to evaluate the significance of associations among categorical variables. A logistic binary regression model was performed to evaluate the independent factors associated with incidence of clinically relevant bleeding. Statistical significance was defined as *p* < 0.05 (two-tailed test, 95% confidence interval). Statistical analyses were performed with SPSS software (SPSS version 20, SPSS Inc., Chicago, IL, USA).

### 2.4. Ethics

This study was conducted in accordance with the guidelines of the Declaration of Helsinki and the principles of good clinical practice. The patients’ data were collected and conserved according to the general data-protection regulation (EU) 2016/679. Institutional review board approval and/or independent ethics committee review was completed for all study sites.

## 3. Results

### 3.1. DOAC-DAA Group

A total of 104 patients were included in the DOAC/DAA group from 12 centers in Italy. Demographic and clinical characteristics are comprehensively reported in Table 1. The cohort was predominantly female (54.8%) with a median age of 80 years. A total of 38 out of 104 patients (36.5%) were cirrhotic with a slightly higher median age compared to non-cirrhotic (82 vs. 79, *p* = 0.023). Among patients with liver cirrhosis, only two (1.9%) had decompensated liver disease (CPT B stage), with one of them showing ascites. The other one patient was in CTP B7 stage for the concomitant presence of hypalbuminemia and increased prothrombin time (expressed as international normalized ratio [INR]) value.

Sofosbuvir/velpatasvir was the most prescribed treatment for both cirrhotic and non-cirrhotic patients (73.7% and 81.8%, respectively). Rivaroxaban was the most common DOAC used in non-cirrhotic patients (40.9%), while in cirrhotics, rivaroxaban and edoxaban were prescribed with the same frequency (26.3%).

As expected, cirrhotic patients had higher median liver stiffness values (17.8 kPa vs. 6.7 kPa) and a higher rate of HAS BLEED score ≥ 3 (63.1% vs. 21.1%) when compared to non-cirrhotic patients.

However, no clinically relevant bleedings were recorded in the cirrhotic group, while only four minor bleedings (3.8%) were recorded among non-cirrhotic patients, thus resulting in no statistically significant difference between the two groups. The management of patients with minor bleeding required an increase in the level of care prompting face-to-face evaluation but did not lead to hospitalization or discontinuation of DOAC or DAA treatment. Among the four patients who experienced clinically relevant bleeding, two were taking concomitant antiplatelet therapy, but the management of these patients did not require a suspension of antiplatelet treatment. Furthermore, each of these patients was taking a different anticoagulant regimen. Additionally, no patients stopped antiviral treatment, although only one non-cirrhotic patient did not achieve SVR 12. No embolic events were recorded in the study population.

### 3.2. Comparison between DOACs/DAAs and VKAs/DAAs Groups

A total of 104 matched patients that switched from DOACs to VKAs before the start of DAA treatment were included in this study (Table 2). The VKA/DAA control group was established by matching the patients by demographic characteristics (age and gender) in a 1:1 ratio with the case group (DOAC/DAA group).

No statistically significant differences emerged between the two group even for other clinical features (e.g., indication for anticoagulation, prevalence of cirrhosis, comorbidities, HAS-BLED score) except for higher INR values at baseline among patients receiving VKAs (2.1 vs. 1.1, respectively).

There were no statistically significant differences between the two groups in the rates of bleeding (major or minor), treatment discontinuation and SVR12 (Table 2 and Table 3), and/or thromboembolic events (Table 4). Similar to the DOAC/DAA group, also in the VKA/DAA group, four cases of minor bleeding were recorded. None of these cases resulted in a withdrawal of antiviral or anticoagulant treatment (Table 3).

On the other hand, one patient (0.9%) one month after starting antiviral therapy experienced major bleeding (peptic ulcer), which resulted in the interruption of both antiviral and VKA treatment as well as the hospitalization of the patient. Although incomplete, the antiviral treatment carried out by this patient led to the SVR-12. Among patients who experienced a bleeding event, none were taking concomitant antiplatelet therapy. Moreover, three of the four patients with minor bleeding were cirrhotic. Table 3 describes features of patients who experienced clinically relevant bleeding.

Although no thromboembolic events occurred in the VKA/DAA control group, when evaluating the baseline INR values, a value of 1.53 was recorded at the 25th percentile, and when assessing individual cases, 44 of 104 patients (42.3%) had an INR value lower than two. A single clinically relevant thrombotic event was recorded in the DOAC group. Specifically, this event was an ischemic stroke that occurred over 3 months after the end of the concomitant antiviral treatment, therefore beyond the observation time (Table 4).

### 3.3. Factors Associated to Bleeding Events

We performed a univariate analysis and a LOGIT binomial analysis with the aim to identify any factors significantly associated to the development of clinically relevant bleeding among patients treated with DAAs and DOACs (Table 5).

The concomitant antiplatelet therapy was the only variable associated with bleeding events among patients on DAAs, with an increased risk of approximately 13.3-fold (1.6–109.051, 95% confidence interval—CI; *p* = 0.016). Moreover, the antiplatelet therapy, evaluated by a LOGIT binomial analysis with demographic characteristics, remained statistically associated with bleeding events, leading to a hazard ratio (HR) of 20-fold higher expression (1.5–287,593 95% CI; *p* = 0.023). However, the analyses performed are at the limits of statistical significance, probably due to the low number of events recorded.

## 4. Discussion

In our real-world multicenter study, we did not observe any major bleedings or thrombotic events during or after DAA and DOAC co-administration. A low incidence of minor bleedings (3.8%) was observed, without the need for discontinuation of either antiviral or anticoagulant treatment. The only clinical variable that increased the risk of bleeding events (HR 13.2) was the concomitant administration to antiplatelet therapy to DOACs, although due to the small number of events, this association is at the limits of statistical significance. In fact, the management of minor bleeding episodes did not require the suspension of antiplatelet treatment. In order to strengthen our results, we compared this population with a group of patients that switched from DOACs to VKAs before the start of antiviral treatment, finding no difference in the incidence of hemorrhagic or thrombotic events.

Few clinical evidences are available on the effect of DOAC and DAA interaction, many of them deriving from clinical trials. Recently, Bellesini et al. [6] re-evaluated the available data within a systematic review. Four studies met the inclusion criteria of the review and three of them observed drug–drug interactions (DDIs) between dabigatran and DAA regimen containing NS3/4A protease inhibitors, showing an increase in the dabigatran concentration of unclear clinical significance. Only one study evaluated DDIs between DAAs and rivaroxaban, apixaban or edoxaban on 54 patients, without reporting serious bleeding events. Indeed, while the DDI between dabigatran and DAAs are known, less evidence has been produced regarding anti-factor Xa drugs, both in terms of bleeding or thromboembolic risks. Dabigatran is a substrate solely of P-gP, while both rivaroxaban, apixaban and edoxaban are also CYP3A4 substrates, undergoing non-negligible hepatic metabolism (50–75%) mediated for 25% by CYP3A4 [7]. Sofosbuvir has no effect on P-gP and CYP3A4, but ledipasvir and velpatasvir exert a mild inhibition on P-gP. In contrast, NS3/4 inhibitors exert a greater inhibition effect on P-gP and also a weak inhibition on CYP3A4. Therefore, we chose to conduct our DDI study only between NS5A/NS5B inhibitors and DOACs.

McDaniel et al. [11] collected a population of 204 patients, including 36 (18%) with cirrhosis, in concomitant treatment with DOACs and DAAs. A low incidence (1.5%) of major bleedings was recorded, without a significant difference in the rate of bleeding events among cirrhotic patients compared to non-cirrhotic subjects. In our cohort, we included 38 cirrhotics (36.5% of total population), including two patients with decompensated liver disease, and no bleeding events were recorded. No statistically significant difference was found in the rate of clinically relevant bleeding compared to the group of non-cirrhotic patients, probably due to the small number of events. Our findings, which include the largest group of cirrhotic patients, seem to suggest that cirrhosis per se might not be a significant risk factor for bleeding during the DOAC and DAA co-administration.

Recently, large multi-national population-based cohort study conducted by Douros et al. [12] on 11,881 patients with non-valvular atrial fibrillation and liver disease (2683 with cirrhosis), it was shown that DOACs were effective and slightly safer than VKAs, as a lower risk of major bleeding was recorded in the DOAC group. In our study, no difference was highlighted in the rate of bleeding when comparing DOAC/DAA and VKA/DAA groups. Furthermore, it can be noted that approximately 42% of patients on VKAs had INR values below the therapeutic target at the beginning of antiviral therapy, thus exposing them to further thrombotic risk. Therefore, we could hypothesize that the DOAC and DAA co-administration, as well as being safe on the risk of bleeding, could potentially be more effective in preventing the thrombotic risk than switching to VKAs.

In our recent study, a hospital-based HCV screening showed a high prevalence of active hepatitis C infection in patients who were HCV-Ab positive and admitted to departments that are not related to liver unit, such as surgery or orthopedics (31–42% of active infection) [13]. In these settings, it is very commons for patient to be treated with DOACs, therefore our findings may be particularly meaningful to help practitioners in the decision regarding concomitant antiviral therapy.

Because the optimal duration of anticoagulation after a symptomatic episode of venous thromboembolism (VTE) is uncertain for deep vein thrombosis (with or without pulmonary embolism), we focused our major outcomes in our cohort on recurrent VTE, other types of thrombosis, bleedings and overall death at three months, as reported in Table 4. When VTE as DVT with or without PE are analyzed at a follow up at 3-6-12 months, the normal rate of recurrence is nearly 3–10% [14,15], while the rate of major bleedings is nearly the same. Yet, the number of events is different when anticoagulation is stopped, and the risk of recurrent VTE increases from 10% (or recurrences in the first year after the end of anticoagulation) to 36% after 10 years [16]. Usually, in similar studies, similar outcomes are also focused at 6 months and 12 months, and these items are focused in another study together to determine changes in liver median stiffness.

The strengths of our study include the large number of cirrhotics under concomitant anticoagulant and antiviral treatment that we collected, as well as the presence of a comparison group on VKAs.

Our findings have the limitation that among 71 cirrhotic patients (DOAC and VKA group), only three patients had decompensated cirrhosis, thus limiting the applicability of our data to the patients at highest bleeding risk. However, DOAC treatment is currently contraindicated in cirrhotic patients with CPT C stage, and furthermore, rivaroxaban administration is not recommended even in CTP B. Thus, the main limitation relies on the retrospective nature of this study, as patients at an increased risk of bleeding may not have been treated simultaneously with DOACs and DAAs.

Despite these limitations, the observation of a low incidence of vascular events in patients treated simultaneously with DOACs and NS5A/NS5B inhibitors, regardless of the severity of liver disease, remains relevant [17]. Furthermore, considering that no difference in vascular events was recorded compared to the VKA group, we believe that our findings can provide important information on how to manage DDIs.

## 5. Conclusions

Our real-word data demonstrated the safety and the efficacy of DOAC and NS5A/NS5B inhibitor coadministration, regardless of the presence of advanced fibrosis. Antiplatelet therapy has been shown to be the only factor capable of significantly increasing the risk of bleeding. No statistically significant difference was highlighted in the risk of embolic and thrombotic events between patients receiving concomitant treatment with DAAs and DOACs and those switching to VKAs. Therefore, replacing DOACs with VKAs does not appear to be of clinical utility; indeed, it potentially exposes the patient to an increase in thrombotic risk due to an inconsistent achievement of the therapeutic target. Finally, our findings underline the safety of DOAC treatment in patients with advanced liver disease, but further studies, including a higher number of patients with liver cirrhosis, are required to confirm these data.

## Figures and Tables

**Figure 1 jcm-13-05807-f001:**
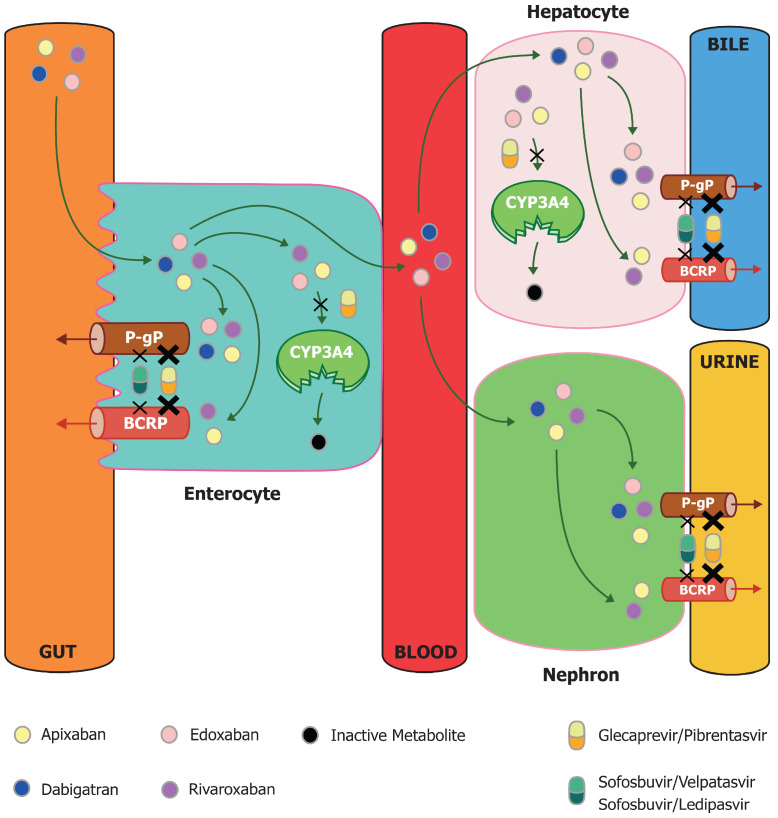
Drug–drug interaction between DOACs and DAAs. DOAC bioavailability is affected by gastrointestinal, renal and hepatic extraction by ATP-binding cassette transporters, such as P-gP and BCRP, as well as drug metabolism by cytochrome P-450 enzymes, such as CYP 3A4. All DOACs are P-gP substrates, while only anti-factor Xa (apixaban, rivaroxaban and edoxaban) are also BCRP and CYP3A4 substrates. NS3/4A inhibitors, such as glecaprevri/pibrentasvir, exert a strong P-gP and BCRP inhibition and a weak CYP 3A4 inhibition. NS5A/NS5B inhibitors exert only a mild P-gP ad BCRP inhibition, without effect on CYP3A4. Abbreviations: BCRP, breast cancer resistance protein; CY3A4, cytochrome P450 3A4; and P-gP, P-glycoprotein.

**Table 1 jcm-13-05807-t001:** Characteristics of pts with and without cirrhosis receiving concomitant DAAs for HCV and DOACs.

	Total	Non-Cirrhotic	Cirrhotic	*p*-Value
N. of pts	104	66	38	
Age, years	80 (71–83)	79 (73–82)	82 (67–83)	0.023
Males	47 (45.2)	29 (43.9)	18 (47.4)	0.735
Indication for anticoagulation				
Atrial Fibrillation	79 (76)	53 (80.3)	26 (68.4)	0.172
Thrombosis	14 (13.5)	10 (15.2)	4 (10.5)	0.259
Other	11 (10.5)	3 (4.5)	8 (21.1)	0.008
DOAC				0.220
Rivaroxaban	37 (35.6)	27 (40.9)	10 (26.3)	
Apixaban	28 (26.9)	19 (28.8)	9 (23.7)	
Edoxaban	19 (18.3)	9 (13.6)	10 (26.3)	
Dabigatran	20 (19.2)	11 (16.7)	9 (23.7)	
Antiplatelet therapy	9 (8.7)	8 (12.1)	1 (2.6)	0.097
Hypertension	67 (64.4)	42 (63.3)	25 (65.8)	0.825
DAAs treatment				0.328
SOF/VEL	82 (78.6)	54 (81.8)	28 (73.7)	
SOF/LDV	22 (21.4)	12 (18.2)	10 (26.3)	
Creatinine at baseline (mg/dL)	0.91 (0.8–1.1)	0.92 (0.9–1.04)	0.91 (0.76–1.11)	0.363
Platelets (10^3^/mL)	177 (132–226)	182 (173–259)	123 (91–174)	0.252
INR	1.1 (1–1.2)	1.1 (1–1.2)	1.2 (1.1–1.3)	0.576
Bilirubin, mg/dL	0.8 (0.58–1)	0.8 (0.5–1)	0.8 (0.7–1.2)	0.745
Albumin, g/dL	4 (3.7–4.3)	4.1 (3.7–4.3)	3.9 (3.4–4.2)	0.435
Liver stiffness median, kPa	9.2 (6.1–16.8)	6.7 (5.3–8.9)	17.8 (15.4–19.6)	<0.001
HAS BLED > 3	38 (36.5)	14 (21.1)	24 (63.1)	<0.001
SVR 12	103 (99)	65 (98.5)	38 (100)	0.446
Clinically relevant bleeding				
Major bleeding	0	0 (0)	0 (0)	-
Minor bleeding	4 (3.8)	4 (6.1)	0 (0)	0.122
DOAC discontinuation	0	0 (0)	0 (0)	-

Median (IQR) for continuous variables and n (percentage) for categorical variables. DAAs: direct antiviral agents; DOAC: direct oral anticoagulants; LDV: Ledipasvir; SOF: Sofosbuvir; SVR 12: sustained virological response at week 12; and VEL: Velpatasvir.

**Table 2 jcm-13-05807-t002:** Characteristics of pts receiving concomitant DAAs for HCV and DOACs or VKAs.

	DOACs/DAAs	VKAs/DAAs	*p*-Value
N. of pts	104	104	
Age, years	80 (71–83)	78 (69–81)	0.079
Males	47 (45.2)	46 (44.2)	0.889
Indication for anticoagulation			0.610
Atrial Fibrillation	79 (76)	83 (79.8)	
Thrombosis	14 (13.5)	14 (13.5)	
Other	11 (10.6)	7 (6.7)	
Antiplatelet therapy	9 (8.7)	14 (13.5)	0.269
Hypertension	67(64.4)	68 (65.4)	0.884
Cirrhosis	38 (36.5)	33 (31.7)	0.465
Decompensation	2 (1.9)	1 (1)	0.138
Esophageal varices	13 (12.6)	9 (8.7)	0.356
DAAs treatment			0.460
SOF/VEL	81(78.6)	86 (82.7)	
SOF/LDV	22 (21.4)	18 (17.3)	
Creatinine at baseline (mg/dL)	0.91 (0.79–1.1)	0.9 (0.77–1.08)	0.873
Platelets (10^3^/mL)	177(132–226)	174 (142–199)	0.244
INR	1.1 (1–1.2)	2.1 (1.53–2.56)	<0.001
Bilirubin, mg/dL	0.8 (0.5–1)	0.9 (0.7–1)	0.389
Albumin, g/dL	4 (3.7–4.3)	3.9 (3.5–4.1)	0.061
Liver stiffness median, kPa	9.2 (6.1–16.8)	9.3 (7.1–14.7)	0.979
HAS BLED score > 3	38 (36.5)	32 (30.8)	0.379
SVR 12	103 (99)	103 (99)	1
Clinically relevant bleeding	4 (3.8)	5 (4.8)	0.316
Major bleeding	0 (0)	1 (0,9)	
Non-major bleeding	4 (3.8)	4 (3.9)	
DOAC or VKA discontinuation	0 (0)	1 (1)	0.316

Median (IQR) for continuous variables and n (percentage) for categorical variables. DAAs: direct antiviral agents; DOAC: direct oral anticoagulants; INR: international normalized ratio; LDV: Ledipasvir; SOF: Sofosbuvir; SVR 12: sustained virological response at week 12; VEL: Velpatasvir; and VKAs: vitamin K antagonists.

**Table 3 jcm-13-05807-t003:** Description of patients who experienced clinically relevant bleeding.

**DOACs/DAAs**												
Pts n.	Age	Sex	Cirrhosis	DOACs	DAAs	Antiplatelet therapy	Creatinine	Platelets (10^3^/mcl)	HAS BLED	DOAC stopped	SVR 12	Bleeding event
1	85	M	No	Dabigatran	SOF/VEL	Aspirin	0.79	141	4	No	Yes	Minor
2	84	F	No	Rivaroxaban	SOF/VEL	no	0.96	112	3	No	Yes	Minor
3	78	F	No	Edoxaban	SOF/LDV	no	1.63	174	3	No	Yes	Minor
4	77	M	No	Apixaban	SOF/LDV	Aspirin	0.62	293	2	No	Yes	Minor
**VKAs/DAAs**												
Pts n.	Age	Sex	Cirrhosis	Warfarin (INR at BL)	DAAs	Antiplatelet therapy	Creatinine	Platelets (10^3^/mcl)	HAS BLED	Warfarin stopped	SVR 12	Bleeding event
1	60	F	No	2.1	SOF/LDV	No	0.7	165	3	No	Yes	Minor
2	40	F	Yes	2.7	SOF/LDV	No	1	115	3	No	Yes	Minor
3	78	F	Yes	2.16	SOF/VEL	No	1.37	160	4	No	Yes	Minor
4	73	F	No	1.95	SOF/LDV	No	0.87	225	2	Yes	Yes	Major
5	69	M	Yes	1.46	SOF/VEL	No	1.1	152	2	No	Yes	Minor

BL: baseline; DAAs: direct antiviral agents; DOAC: direct oral anticoagulants; INR: international normalized ratio; LDV: Ledipasvir; SOF: Sofosbuvir; SVR 12: sustained virological response at week 12; VEL: Velpatasvir; and VKAs: vitamin K antagonists.

**Table 4 jcm-13-05807-t004:** Vascular events (haemorrhagic or thrombotic) in DOAC/DAA and VKA/DAA groups.

	DOACs	VKAs	*p*-Value
N. of pts	104	104	
Clinically relevant bleeding	4 (3.8)	5 (4.8)	0.316
Major bleeding	0 (0)	1 (0.9)	
Non-major bleeding	4 (3.8)	4 (3.9)	
Clinically relevant thrombotic event	1 (0.9)	0 (0)	0.248
Arterial thromboembolism	1 * (0.9)	0	0.248
Venous thromboembolism	0	0	
Venous thrombosis	0	0	

Median (IQR) for continuous variables and *n* (percentage) for categorical variables. DAAs: direct antiviral agents; DOAC: direct oral anticoagulants; and VKAs: vitamin K antagonists. * event occurred more than 3 months after the end of antiviral therapy.

**Table 5 jcm-13-05807-t005:** Factors associated with incidence of clinically relevant bleeding.

Variables	Univariate Analysis		Multivariate Analysis	
	HR (95%CI)	*p*-Value	HR (95%CI)	*p*-Value
Age, year	1.071 (0.935–1.226)	0.324		-
Male gender	1.222 (0.166–9.024)	0.844		-
LSM, kPa	0.923 (0.757–1.126)	0.431		-
Platelets, (10^3^/mL)	0.999 (0.986–1.012)	0.863		-
INR	0.809 (0.002–280.727)	0.943		-
HAS BLED > 3	5.571 (0.558–55.580)	0.143		-
Antiplatelet therapy	13.286 (1.619–109.051)	0.016	20.815 (1.506–287.593)	0.023

HR: hazard ratio; LSM: liver stiffness measurement.

## Data Availability

The data presented in this study are available upon request from the corresponding author due to privacy, legal and ethical reasons.

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
