# Peer review of "Safety of Sofosbuvir-Based Direct-Acting Antivirals for Hepatitis C Virus Infection and Direct Oral Anticoagulant Co-Administration"

_jcm, 2024, doi:10.3390/jcm13195807_

Round 1

Reviewer 1 Report

Comments and Suggestions for Authors

I have reviewed the manuscript entitled ‘Safety of sofosbuvir-based direct-acting antivirals for Hepatitis C Virus infection and direct oral anticoagulants co-administration.

The manuscript addressed the use of DOACs with antiviral agent sofosbuvir.

The abstract is fluent however the patient selection should be emphasized as patients with atrial fibrillation.

In conclusion, the authors mentioned the role of DOAC use in other contraindications if possible data is available.

Comments on the Quality of English Language

Minor editing is required.

Author Response

I have reviewed the manuscript entitled ‘Safety of sofosbuvir-based direct-acting antivirals for Hepatitis C Virus infection and direct oral anticoagulants co-administration.

The manuscript addressed the use of DOACs with antiviral agent sofosbuvir.

The abstract is fluent however the patient selection should be emphasized as patients with atrial fibrillation.

In conclusion, the authors mentioned the role of DOAC use in other contraindications if possible data is available.

We thank the reviewer and agree with his comments. We have revised the text to emphasize patient selection.

Reviewer 2 Report

Comments and Suggestions for Authors

It is an interesting manuscript about “Safety of sofosbuvir-based direct-acting antivirals for Hepatitis C Virus infection and direct oral anticoagulants co-administration.”.

*My concern is determined in the following points.

The present real-word data demonstrated the safety and the efficacy of DOAC and NS5A/NS5B inhibitors coadministration, regardless of the presence of cirrhosis. Antiplatelet therapy has been shown to be the only factor capable of significantly increasing the risk of bleeding. No statistically significant difference was highlighted in the risk of embolic and thrombotic events between patients receiving concomitant treatment with DAAs and DOACs and those switching to VKAs. Therefore, replacing DOACs with VKAs does not appear to be of clinical utility, indeed it potentially exposes the patient to an increase in thrombotic risk due to an inconsistent achievement of the therapeutic target. Finally, the present findings underline the safety of DOAC treatment in patients with advanced liver disease, but further studies are required to confirm these data.

*An increased incidence of atherothrombotic events [e.g. coronary artery disease (CAD), atrial fibrillation (AF)] has been observed in HCV seropositive patients. On the other hand, an increased bleeding risk is another clinical issue, particularly in subjects with liver cirrhosis, gastroesophageal varices, portal hypertension, thrombocytopenia and alcohol consumption. The introduction and progressively greater use of direct-acting antivirals (DAAs) during the last decade has enabled a sustained virological response to be achieved in a significant percentage of patients. However, due to the high cardiovascular risk profile in HCV-infected patients, the concomitant use of antithrombotic therapies is often required, bearing in mind the possible contraindications. For example, despite better pharmacokinetic and pharmacodynamic properties compared with vitamin K-antagonists, plasma level fluctuations of direct oral anticoagulants (DOACs) due to pathological conditions (e.g. chronic kidney diseases or hepatic cirrhosis) or drug-drug interactions (DDIs) may be of great importance as regards their safety profile and overall clinical benefit. The significant DDIs were observed between antithrombotic and HCV antiviral drugs.

*Above mentioned should be referred to.

Author Response

It is an interesting manuscript about “Safety of sofosbuvir-based direct-acting antivirals for Hepatitis C Virus infection and direct oral anticoagulants co-administration.”.

*My concern is determined in the following points.

Response: 

We thank the reviewer for his reports. We have modified the text to make it more explanatory by adding references that support our statements.

The present real-word data demonstrated the safety and the efficacy of DOAC and NS5A/NS5B inhibitors coadministration, regardless of the presence of cirrhosis. Antiplatelet therapy has been shown to be the only factor capable of significantly increasing the risk of bleeding. No statistically significant difference was highlighted in the risk of embolic and thrombotic events between patients receiving concomitant treatment with DAAs and DOACs and those switching to VKAs. Therefore, replacing DOACs with VKAs does not appear to be of clinical utility, indeed it potentially exposes the patient to an increase in thrombotic risk due to an inconsistent achievement of the therapeutic target. Finally, the present findings underline the safety of DOAC treatment in patients with advanced liver disease, but further studies are required to confirm these data.

*An increased incidence of atherothrombotic events [e.g. coronary artery disease (CAD), atrial fibrillation (AF)] has been observed in HCV seropositive patients. On the other hand, an increased bleeding risk is another clinical issue, particularly in subjects with liver cirrhosis, gastroesophageal varices, portal hypertension, thrombocytopenia and alcohol consumption. The introduction and progressively greater use of direct-acting antivirals (DAAs) during the last decade has enabled a sustained virological response to be achieved in a significant percentage of patients. However, due to the high cardiovascular risk profile in HCV-infected patients, the concomitant use of antithrombotic therapies is often required, bearing in mind the possible contraindications. For example, despite better pharmacokinetic and pharmacodynamic properties compared with vitamin K-antagonists, plasma level fluctuations of direct oral anticoagulants (DOACs) due to pathological conditions (e.g. chronic kidney diseases or hepatic cirrhosis) or drug-drug interactions (DDIs) may be of great importance as regards their safety profile and overall clinical benefit. The significant DDIs were observed between antithrombotic and HCV antiviral drugs.

*Above mentioned should be referred to.

Reviewer 3 Report

Comments and Suggestions for Authors

The article is a very interesting work, which describes very well the opportunity of using DOAC in the combination with direct acting antivirals. The article is accepted in current form.

The article was a very interesting one, which describes the safety of direct acting antivirals with the co-administration of DOAC.
The topic is very original, there is few data in the literature regarding this subject.
The methodology is very well written.
The conclusion are supported by the results.
The references are appropriate for the subject.

Comments on the Quality of English Language

The quality of English is very well.

Author Response

The article is a very interesting work, which describes very well the opportunity of using DOAC in the combination with direct acting antivirals. The article is accepted in current form.

The article was a very interesting one, which describes the safety of direct acting antivirals with the co-administration of DOAC.
The topic is very original, there is few data in the literature regarding this subject.
The methodology is very well written.
The conclusion are supported by the results.
The references are appropriate for the subject.

Response:

We thank the reviewer for his considerations on the manuscript. We are pleased to have met the journal's expectations.

Reviewer 4 Report

Comments and Suggestions for Authors

Rosato et al performed a real-world multicenter study and found that the bleedings incidence in patient that were on DAAs and DOACs co-administration was low and comparable to those patients switched to DAA/VKS co-therapy. This observation is the largest of its kind and of significant clinical value.

Minor comments only:

1.     Is there any data on the Quantitative or Qualitative measures of DOAC blood level or anticoagulant activity in the DOAC/DAA patients? This might help to understand the effect of different DAAs on the plasma level or subsequent anticoagulant effect of different DOACs.

2.     The conclusion should be more careful: the observations were made in a group of non-cirrhotic patients and cirrhotic patients that were mostly in a compensated state.

3.     Maybe discuss the rationale behind the antiplatelet therapy and whether the termination of antiplatelet therapy improves the bleeding problem.

4.     Typos: line 339 “rated”; line 96 “edoxaban”.

Author Response

Rosato et al performed a real-world multicenter study and found that the bleedings incidence in patient that were on DAAs and DOACs co-administration was low and comparable to those patients switched to DAA/VKS co-therapy. This observation is the largest of its kind and of significant clinical value.

Minor comments only:

Is there any data on the Quantitative or Qualitative measures of DOAC blood level or anticoagulant activity in the DOAC/DAA patients? This might help to understand the effect of different DAAs on the plasma level or subsequent anticoagulant effect of different DOACs.

Response:

We thank the reviewer for pointing this out. Unfortunately, we do not have data on the plasma values ​​of DOACs. Baseline INR values ​​were collected and showed no significant alterations and similarly no further alterations were reported in patients who had minor bleeding events. The reviewer's assessment is highly worthy of interest and could be the subject of further investigations.

Comment 2: 

The conclusion should be more careful: the observations were made in a group of non-cirrhotic patients and cirrhotic patients that were mostly in a compensated state.

Response 2: 

We agree with the reviewer and have modified some statements, making the conclusions more cautious.

Comment 3: 

Maybe discuss the rationale behind the antiplatelet therapy and whether the termination of antiplatelet therapy improves the bleeding problem.

Response 3: 

We thank the reviewer for the reports. In our study population, antiplatelet therapy was the only variable associated with a higher rate of minor bleeding. Only 4 cases of minor bleeding were recorded, therefore, as described in the results and discussion, this association is borderline statistically significant, but we have reported this data for completeness and transparency. We understand the reviewer's concern about the possible suspension of antiplatelet therapy and have clarified in the text that no therapeutic modification was necessary following the minor bleeding episodes.

Comment 4:

Typos: line 339 “rated”; line 96 “edoxaban”.

Response 4:

We thank the reviewer for the reports. We have made the corrections.